# Anticoagulation in Atrial Fibrillation Cardioversion: What Is Crucial to Take into Account

**DOI:** 10.3390/jcm10153212

**Published:** 2021-07-21

**Authors:** Fabiana Lucà, Simona Giubilato, Stefania Angela Di Fusco, Laura Piccioni, Carmelo Massimiliano Rao, Annamaria Iorio, Laura Cipolletta, Emilia D’Elia, Sandro Gelsomino, Roberta Rossini, Furio Colivicchi, Michele Massimo Gulizia

**Affiliations:** 1Division of Cardiology, Big Metropolitan Hospital, Bianchi Melacrino Morelli, 89129 Reggio Calabria, Italy; massimo.rao@libero.it; 2Division of Cardiology, Cannizzaro Hospital, 95121 Catania, Italy; simogiub@hotmail.com; 3Division of Cardiology, S. Filippo Neri Hospital, 00135 Roma, Italy; doctstefania@hotmail.com (S.A.D.F.); furio.colivicchi@gmail.com (F.C.); 4Division of Cardiology, Cardiovascular Departiment, Civile Giuseppe Mazzini Hospital, 64100 Teramo, Italy; 5Division of Cardiology, Papa Giovanni XXIII Hospital, 24127 Bergamo, Italy; anita.iorio@hotmail.it (A.I.); edelia@asst-pg23.it (E.D.); 6Division of Cardiology, Department of Cardiovascular Sciences, Ancona University Hospital, 60126 Ancona, Italy; cipollettalaura@gmail.com; 7Cardiothoracic Department, Maastricht University Hospital, 6202 AZ Maastricht, The Netherlands; sandro.gelsomino@gmail.com; 8Division of Cardiology, S. Croce e Carle Hospital, 12100 Cuneo, Italy; roberta.rossini2@gmail.com; 9Division of Cardiology, Garibaldi-Nesima Hospital, 95123 Catania, Italy; michele.gulizia60@gmail.com; 10Heart Care Foundation, 50121 Florence, Italy

**Keywords:** atrial fibrillation, electrical cardioversion, pharmacological cardioversion, non-vitamin K antagonist oral anticoagulants

## Abstract

The therapeutic dilemma between rhythm and rate control in the management of atrial fibrillation (AF) is still unresolved and electrical or pharmacological cardioversion (CV) frequently represents a useful strategy. The most recent guidelines recommend anticoagulation according to individual thromboembolic risk. Vitamin K antagonists (VKAs) have been routinely used to prevent thromboembolic events. Non-vitamin K antagonist oral anticoagulants (NOACs) represent a significant advance due to their more predictable therapeutic effect and more favorable hemorrhagic risk profile. In hemodynamically unstable patients, an emergency electrical cardioversion (ECV) must be performed. In this situation, intravenous heparin or low molecular weight heparin (LMWH) should be administered before CV. In patients with AF occurring within less than 48 h, synchronized direct ECV should be the elective procedure, as it restores sinus rhythm quicker and more successfully than pharmacological cardioversion (PCV) and is associated with shorter length of hospitalization. Patients with acute onset AF were traditionally considered at lower risk of thromboembolic events due to the shorter time for atrial thrombus formation. In patients with hemodynamic stability and AF for more than 48 h, an ECV should be planned after at least 3 weeks of anticoagulation therapy. Alternatively, transesophageal echocardiography (TEE) to rule out left atrial appendage thrombus (LAAT) should be performed, followed by ECV and anticoagulation for at least 4 weeks. Theoretically, the standardized use of TEE before CV allows a better stratification of thromboembolic risk, although data available to date are not univocal.

## 1. Introduction

Atrial fibrillation (AF) patient care encompasses different possible management strategies which are classified as rhythm-control therapies, aimed at restoring and maintaining the sinus rhythm, and rate-control therapies, aimed at ensuring an appropriate control of heart rate during AF. Although a rhythm-control strategy may have some clinical benefit, it does not seem to offer advantages in terms of mortality or morbidity over a rate-control strategy [1]. Although some observational data [2,3] is in favor of rhythm control strategy, in order to reduce thromboembolic risk in patients with paroxysmal in contrast with persistent AF (due to a reduced risk of stroke in patients with paroxysmal as opposed to persistent AF), randomized clinical trials (RCTs), including the Atrial Fibrillation Follow-up Investigation of Rhythm Management trial, have failed to show significant difference in survival or thromboembolic events related to rhythm control compared to rate control strategy [1]. Notably, thromboembolic risk needs to be evaluated according to the CHA2DS2-VASc score (congestive heart failure, hypertension (1 point for presence of each), age ≥ 75 years (2 points), diabetes mellitus (1 point), stroke/TIA (2 points), vascular disease, age 65 to 74 years, female sex (1 point for presence of each); scores range from 0 to 9), regardless of the adopted strategy.

Nonetheless, rhythm control still represents the preferred strategy, especially in young patients who are symptomatic and in patients with hemodynamic instability. 

## 2. Acute Hemodynamic Instability 

Acute hemodynamic instability in AF, due to a rapid ventricular rate (typically >150 bpm or higher in patients compromised by co-morbidities), is characterized by clinical manifestations such as syncope, acute pulmonary edema, myocardial ischemia, symptomatic hypotension, or cardiogenic shock. In those patients an emergency electrical cardioversion has to be promptly performed, and anticoagulation should be started as soon as possible [4].

Electrical or pharmacological cardioversion (CV) is the cornerstone of the rhythm management strategy. Electrical cardioversion (ECV) is not only recommended in hemodynamically unstable patients but is also the preferred strategy in stable patients when AF duration is prolonged, whereas pharmacological cardioversion (PCV) may be preferred in recent onset AF [4]. In stable patients, pharmacological and electrical cardioversion can both be suitable options to perform. Electrical cardioversion is more effective, although sedation is needed [4]. Of note, pre-treatment with AADs can improve the efficacy of elective electrical cardioversion. Irrespective of the CV method employed, reestablishment of sinus rhythm confers a substantial possibility of peri-cardioversion thromboembolism, with a stroke rate enclosed by 5 and 7% in non-anticoagulated patients [1,2,3,4,5]. The risk of stroke or systemic embolism (SSE) is higher forthwith next to CV, with 82% of cases occurring within 72 h and 98% of accidents manifesting within 10 days [4]. This risk is related to the potential embolization of previous thrombus from the atrial appendage after recovery of effective atrial contractility. Furthermore, the mechanism of CV might support novel thrombus development as a result of transitory atrial stunning. For these reasons, current European Society of Cardiology (ESC) guidelines recommend not less than 3 weeks of adequate anticoagulation before CV, accompanied by minimum of 4 weeks of anticoagulation after the procedure in patients with AF duration > 48 h (or unknown) irrespective of CHA2DS2-VASc score or transesophageal echocardiography (TEE) done to exclude left atrial thrombi [4].

In the last 50 years, a great deal of clinical experience has been accumulated on the use of VKAs for anticoagulation in AF. However, this treatment has never been validated in large RCT. To prevent thromboembolic complications, the availability of non-vitamin K antagonist oral anticoagulants (NOACs) represents a big step forward thanks to their more predictable therapeutic effect and more favorable hemorrhagic risk profile [6]. The advantage of a rapid and predictable action onset of these drugs is particularly useful in the peri-cardioversion setting. Indeed, in this specific setting the longer and variable time needed to achieve an effective anticoagulation with VKAs often requires the use of a heparin bridge and a delayed CV. NOACs’ use in AF cardioversion is epidemiologically relevant in clinical practice [7,8].

In the present paper we review NOACs’ use in AF patients undergoing CV, with a focus on safety and efficacy findings. Furthermore, on the basis of current evidence we share pragmatic thoughts about NOAC using in different peri-cardioversion scenarios. As far as we are concerned, the most recent ESC guidelines should be considered a useful guide in order to avoid fraught roads with danger and tricky situations as has been summarized in Figure 1.

## 3. Electrical Cardioversion in Emergency

Recent-onset episodes of AF are among the most usual arrhythmias that physicians deal with, constituting almost 35% of hospital accesses due to arrhythmias [9]. The prevalence of AF is enhancing owing to the increase of the age of population. Despite an almost stable relative rate of hospitalization, the emergency department admissions for AF are increasing [10].

The first step in the management of AF patients consists to assess if the patient is hemodynamically stable or is presenting symptoms. In hemodynamically unstable AF patients, an emergency ECV should be carried out promptly. The steps to follow for emergency management of AF are prevention of thromboembolism and hemodynamic stabilization, followed by symptom relief. Synchronized direct current ECV is the favored practice in severely hemodynamically unstable patients with hypotension, acute coronary syndrome, or pulmonary edema due to new-onset AF [11]. Although ECV is connected with a high initial success percentage (68–98%), and it can be effective for solving an acute compromised situation [12], it is not a risk-free procedure. Moreover, long-term maintenance of sinus rhythm is not guaranteed, and a relapse of AF after ECV is associated with a poor outcome [13]. As highlighted in the more recent guidelines for the management of AF, defibrillator devices with biphasic waveforms are more effective than monophasic ones, and an anterior–posterior electrode position should be preferred over antero–lateral, as the ECV procedure seems to be safer and more successful [4].

Moreover, maximum fixed-energy electrical CV was more effective than an energy-escalation strategy in achieving sinus rhythm based on current data. It has been claimed that an initial synchronized shock at maximum defibrillator output (360 J) was a reasonable approach without an increasing in adverse events [14]. What is particularly noticeable is that an initial energy setting of 360 J resulted more efficiently than traditional approach, particularly when the duration of AF is longer [15].

Patients with the Wolff–Parkinson–White syndrome suffering from acute AF require special precautions in their management. Due to the prolonged atrial conduction WPW, characterized by a longer maximal atrial conduction delay and wider conduction delay zone, the atrial vulnerability to develop of AF is greater [16]. Ventricular fibrillation could be caused by rapid atrioventricular conduction over the accessory conduction pathway. Therefore, in a patient with Wolff–Parkinson–White syndrome, drugs that block atrioventricular node conduction (digoxin, β-blockers, and calcium channel blockers) are not indicated since they are ineffective on the accessory pathways so they could trigger ventricular fibrillation. In emergency setting immediate CV should not be delayed in order to perform adequate anticoagulation. In this situation, intravenous heparin or low molecular weight heparin (LMWH) should be administered before CV [17].

After cardioversion a long term OAC strategy (OAC for all patients if a CHAD2DS2VASc ≥ 1_men_ or ≥2_female_) or short term OAC strategy (4 weeks OAC if a CHAD2DS2VASc = 0_men_ or ≥1_female_ (optional if AF onset < 24 h) is needed [4].

### 3.1. Electrical Cardioversion in Patients with AF Which Occurred within Less Than 48 h

In the absence of hemodynamic compromise, the management of AF is guided by symptoms and its duration. Synchronized direct ECV should be the elective procedure to be followed by physicians, as it re-establishes faster and more efficiently than PCV, and involves a briefer hospitalization [18]. Of note, pre-treatment with antiarrhythmic drugs can improve the efficacy of ECV.

Recently, the RACE7 ACWAS trial [19], demonstrated that a wait-and-watch approach with rate control medication only and CV within 48 h of symptom onset was as safe as, and non-inferior to, immediate CV of paroxysmal AF, which is likely to terminate spontaneously within 24 h.

In clinical practice, there is clear evidence of a lack regarding the need for anticoagulation when AF has occurred within 48 h in naïve patients with a very low risk of stroke. AF itself independently increases stroke risk by 5-fold, which represents one of the leading causes of morbidity and death in AF patients [20]. The annual risk of stroke in those patients is greater than 20%. It is important to recognize that in absence of additional clinical stroke risk, assessed by CHA_2_DS_2_-VASc score, AF patients without anticoagulant therapy have an ischemic stroke rate of 0.43% per year. Conversely in those with 1 additional point in the CHA_2_DS_2_-VASc score the risk range from 1.18% to 3.50% per year. Another commonly held claim is that an increased risk of stroke and systemic thromboembolism in AF is, in a certain way, linked to a persistent prothrombotic state, as demonstrated by the increasing of platelet activation, thrombin formation, and inflammation with a reduction fibrinolysis process and by the endothelial dysfunction both. Unsurprisingly, what is particularly noticeable is that even young and very low-risk patients with AF have prothrombotic alterations, and the so-called prothrombotic fibrin clot phenotype could be often present. Głowicki and coworkers showed that a prothrombotic pathway might involve an increasing risk among patients with AF with the CHA2DS2-VASc score of 1 despite the sex [21].

Patients with acute onset AF were traditionally considered at lower risk of thrombo-embolic events, related to the shorter time for atrial thrombus formation [22]. However, CV has an inherent risk of stroke in non-anticoagulated patients, which is lowered by anticoagulant drugs administration [4]. Nonetheless, those patients with new onset AF and stroke risk factors take probably more advantages of using oral anticoagulant (OAC). The most common risk stratification score currently utilized is the CHA_2_DS_2_-VASc [23]. A CHA_2_DS_2_-VASc score of 1 or more for men, and 2 or more for women is considered the cut off value for starting oral anticoagulation, in order to protect patients from stroke events. The incidence of stroke and thrombo-embolic events varies significantly in patients with CHA_2_DS_2_-VASc scores of 1 or 2. Of note, an age of more than 65 years is associated with an increased risk of stroke, and it also potentiates other risk factors, such as heart failure and sex [24].

Regarding the decision whether to start an OAC for a new-onset AF in naïve patients, several studies showed that starting pre-cardioversion anticoagulation in all patients with AF episodes of less than 24 h, or even 12 h, would afford even greater safety [25,26]. In the Fin CV (Finnish Cardioversion) Study, a large multicenter retrospective cohort trial exploring the occurrence and risk factors of thromboembolic complications after CV in acute AF [27], 7660 cardioversions were carried out in 3143 consecutive patients with AF that occurred within 48 h. The overwhelming preponderance (88%) of CVs were ECVs. Embolic complications at 30 days were reported after 5116 effective CVs in 2481 patients with neither oral anticoagulation nor peri-procedural heparin therapy. Thirty-eight embolic events occurred (0.7% of efficacious procedures; 95% confidence interval (CI): 0.5% to 1.0%) after a follow-up of 1 month, and 31 of these were strokes. Logistic regression analyses revealed that age, female sex, heart failure, and diabetes were independent predictors of embolic events. Moreover, with the coexistence of multiple risk factors, the risk results in an extremely elevated risk (approximately 10%), that is notably higher than after elective CV of AF with conventional anticoagulation. Based upon these results, the authors affirmed that for certain subgroups of patients with a new onset AF, the risk of a stroke becomes notable and a pre/post CV anticoagulant approach is preferable. Indeed, according to the more recent guidelines for managing AF, a pre-cardioversion anticoagulation therapy is now recommended for all patients, irrespective of risk factors for stroke (Class IIa B). The same thing is valid for the four-week post-procedural anticoagulant therapy, both in case of PCV or ECV [4].

### 3.2. Electrical Cardioversion in AF Which Lasted for Longer Than 48 h

In patients with hemodynamic stability and an AF which lasted for longer than 48 h or of unknown time of onset, ECV should be planned after an appropriate anticoagulation therapy, either with 4 consecutive weeks of warfarin with weekly therapeutic INR (2–3) or 4 weeks of NOACs without any interruption. Alternatively, a TEE to exclude left atrial appendage thrombus (LAAT) could also be performed followed by an immediate ECV and a subsequent oral anticoagulation for at least 4 weeks [28]. A window time of 48 h of AF is generally considered the boundary beyond which LAAT may organize, and CV-caused atrial stunning is likely to occur [29]. As previously reported, thromboembolic events are also frequent in the first month after CV [27].

Although it could seem generally better to reinstate a sinus rhythm in all patients with persistent AF, all studies that have evaluated rhythm control in comparison with rate control (with appropriate anticoagulation) have resulted in neutral outcomes [30,31].

Factors which might facilitate an attempt at rhythm control should be evaluated [4].

The left atrial volume index (LAVI) could represent an effective marker in predicting the maintenance of SR contributing to the identify patients in which CV is likely to be successful. A cut-off 55 mL/m^2^ has been proposed [32]. Another factor which should be taken into account is the left atrial diameter since it seems to be associated with AF recurrence after cardioversion if it is larger than 44 mm [33]. Conversely, it could be that a heterogeneous electrical activation of the LA appears to be related to AF recurrence. An advanced interatrial block (aIAB), P wave duration > 120 ms, and biphasic P waves in the inferior leads have been recognized as independent predictors of AF recurrence [34].

However, in the case of undated AF, not only an adequate anticoagulation, but also proper control of the heart rate is required. Beta-blockers, digoxin, diltiazem, or verapamil are recommended to control heart rate in AF patients with left ventricle ejection fraction (LVEF) ≥ 40%, while calcium antagonists should be avoided in the case of left ventricular dysfunction.

Lastly, when antiarrhythmic drug therapy is used to preserve sinus rhythm after ECV, oral administration of amiodarone (a few weeks) or class IC antiarrhythmic drugs in order to improve the chance of a successful procedure should be considered, together with a proper anticoagulant [35,36].

## 4. TEE Guided Approach

TEE is a useful tool to identify thrombi in the atrial cavities and in the left appendage, which are the main sites of thrombotic formation [37]. Not only atrial thrombosis, but also the presence of spontaneous echo contrast known as “smoke effect” described as a swirling pattern of increased echogenicity, and decreased atrial emptying velocities (peak LAA velocity < 20 cm/s), are associated with a greater risk of stroke or peripheral embolism. It is important to recognize that also LA dilation, reduced LA strain, LAA thrombus, and non-chicken wing LAA morphology have been proposed as predictive factors of stroke [38]. Likewise, the presence of normal velocities in appendage (normal values 20–40 cm/s) [39] seems to be associated with a successful ECV and the maintenance of long-term sinus rhythm [40], and a mean LAA peak emptying velocity peak > 40 cm/s has proved to be an independent predictor of no AF recurrence after one-year [40].

The ACUTE (Assessment of Cardioversion Using Transesophageal Echocardiography) is the only randomized prospective multicenter trial comparing a conventional strategy of CV following oral anticoagulation for at least 3 weeks with a strategy TEE-guided in 1222 patients with AF for over 48 h [41]. When the TEE did not document the presence of thrombotic formations, anticoagulation with heparin and subsequent CV accompanied by at least 4 weeks of oral anticoagulation were performed. On the contrary, early identification of atrial thrombosis required re-evaluation with a TEE after at least 3 weeks of anticoagulation. The primary end-point of peripheral embolism and/or stroke at 8 weeks post CV did not show significant differences between the two strategies (3 events, 0.5%, in the group undergoing conventional therapy, n = 603 vs. 5 events, 0.8% or in the group subjected to TEE, n = 619, *p* = 0.5). In addition, even the success rate 8 weeks after CV did not differ between the two groups (80.3% and 52.7% in the TEE-guided strategy vs. 52.7% and 50.4% in the conventional strategy, respectively, *p* = NS). However, in the TEE arm a reduction of hemorrhagic events was observed (cumulative of major and minor events 2.9 vs. 5.5%; *p* = 0.03), without remarkable differences for major bleedings (0.8% vs. 1.5%, *p* = 0.26).

However, the ACUTE trial was stopped before reaching the sample size of 3000 patients due to the interim analysis showing a lower percentage of embolic events than expected, and due to problems in recruitment. Therefore, the study did not reach the statistical power necessary to highlight significant differences between the two groups [41].

A TEE-guided approach allowed to obtain a shorter duration of the oral anticoagulation, and consequently a global reduction of hemorrhagic events in this arm. Gallagher et al. focused on the importance of a proper anticoagulation in case of a conventional strategy: if oral anticoagulation therapy is conducted correctly maintaining an INR > 2.4, the risk of embolic events is reduced compared to patients with an INR < 2.4 (0% vs. 0.93%, *p* = 0.012) [42].

Lastly, both strategies showed an acceptable and substantially overlapping safety profile.

It should be noticed that a negative TEE does not exclude the possibility of a thromboembolic event after the procedure. This is due to the fact that the TEE method is not free of false negatives, and also that a non-negligible number of thromboembolic events after CV are caused by migration of thrombi generated after the procedure, and not preexisting Figure 2.

It is in light of this evidence that the most recent guidelines for AF management recommend continuing oral anticoagulation for at least 4 weeks after CV, both in the case of a TEE guided approach, and in case of a conventional therapy [4].

Furthermore, the increasing use of it has raised concern about the monitoring of anticoagulation compliance before and after CV. The routine TEE-guided approach guarantees the safety of CV if patients take NOACs. However, a recent prospective study on 311 patients shows that a verbal assessment of drug compliance using a standardized questionnaire maintains the same safety as a TEE-guided approach [43].

## 5. NOACs in the Setting of Cardioversion

For several decades VKAs have been routinely used to prevent thromboembolic events in various clinical settings. Despite never having been validated in large RCTs, available data shows that VKAs during peri-cardioversion decreases the peri-procedural occurrence of thromboembolic complications by up to 1.6% [44].

NOACs are being increasingly employed in routine clinical practice due to their favorable benefit–risk profile compared with VKAs. Moreover, the chance to reach a quicker and more durable control of anticoagulation in patients requiring CV is beneficial in terms of time to procedure and patients’ safety. Conversely, the mean time needed to achieve an effective anticoagulation with VKAs is long and variable and it is often necessary to use LMWH as bridge in order to avoid postponing scheduled CV.

Most of the information regarding the efficacy and safety of NOACs in CV comes from subgroup analysis of the major trials on NOACs in nonvalvular atrial fibrillation (RELY, ROCKET AF, ARISTOTLE, ENGAGE AF-TIMI 48) [45,46,47,48] (Table 1).

However, despite the low event rate shown in these studies in patients undergoing CV treated with NOACs, the most important limitation of these studies was the long period of anticoagulant treatment preceding CV. In the RELY (randomized evaluation of long-term anticoagulant therapy: dabigatran vs. warfarin) sub-study, two-thirds of patients were treated for more than 3 weeks with dabigatran and one in four of patients had TEE performed on them. In the ARISTOTLE (Apixaban for Reduction in Stroke and Other Thromboembolic Events in Atrial Fibrillation) sub-study, the mean time from study entry to first CV for patients assigned to apixaban was 251 ± 248 days. Thus, on the basis of these data, there would be no justification for switching a NOAC to VKA to perform an elective CV in the setting of AF recurrence in patients chronically treated with NOACs. More recently, dedicated prospective trials have been conducted with patients undergoing CV comparing VKA to rivaroxaban [49], edoxaban [50], and apixaban [51]. These trials have enrolled thousands of patients who were scheduled for rapid or delayed CV (Table 2). Although none of these trials had the statistical power to demonstrate superiority or non-inferiority of treatment with NOAC vs. VKA, all studies reported a lower rate of thrombotic and bleeding events in both NOACs-treated and VKA-treated patients (statistically non-significant differences), except for lower ischemic events with apixaban in the EMANATE trial.

## 6. Dabigatran

A post hoc analysis of the RE-LY trial compared dabigatran to warfarin in the post-cardioversion setting. This study evaluated 1270 patients (7% of the 18,113 patients enrolled) who underwent PCV or ECV an allocated to dabigatran or warfarin for NVAF [45]. Both doses of dabigatran (110 and 150 mg daily) were compared with warfarin. A pre-cardioversion TEE was performed in 25.5%, 24.1%, and 13.3% for the dabigatran 110 mg, dabigatran 150 mg, and warfarin arms, respectively. No significant differences between three groups were found in the incidence of LAAT. Moreover, the rate of embolic events (stroke and systemic embolism) was similar regardless of the strategy of CV (TEE- vs. non-TEE-guided approach). For dabigatran 110 mg, dabigatran 150 mg, and warfarin, the rate of embolic events within 30 days of CV were 0.8%, 0.3%, and 0.6%, respectively; dabigatran 110 mg vs. warfarin, *p* = 0.71; dabigatran 150 mg vs. warfarin, *p* = 0.40. Major bleeding within 30 days of CV occurred in 1.7%, 0.6%, and 0.6%, respectively; dabigatran 110 mg vs. warfarin, *p* = 0.06; dabigatran 150 mg vs. warfarin, *p* = 0.99.

### 6.1. Rivaroxaban

A post hoc analysis of the ROCKET AF (Rivaroxaban Once-daily Oral Direct Factor Xa Inhibition Compared with Vitamin K Antagonism for Prevention of Stroke and Embolism Trial in Atrial Fibrillation) study evaluated outcomes associated with both CV (n = 375 48.2% ECV, 51.8% PCV) and catheter ablation (n = 85) procedures [46] demonstrating that the incidence of embolic events and major or non-major bleeding in the rivaroxaban and warfarin groups were similar. However, this analysis reported only composite CV and ablation results and data regarding the use of pre-cardioversion TEE were not reported.

X-VeRT (eXplore the efficacy and safety of once-daily oral riVaroxaban for the prevention of caRdiovascular events in patients with non-valvular aTrial fibrillation scheduled for cardioversion) was the first prospective randomized trial designed to explore the efficacy and safety of once-daily rivaroxaban compared with dose-adjusted VKA treatment (with or without heparin), in patients undergoing elective CV either TEE- or non–TEE-guided. This study randomized 1504 patients with NVAF lasting ≥48 h. Patients were randomized 2:1 to receive rivaroxaban 20 mg daily (or 15 mg daily with CrCl 30–49 mL/min) or warfarin (target INR 2–3). Heparin bridging was used at the discretion of the investigator. Forty-three percent of patients were experienced with OAC, defined as ≥6 weeks of oral anticoagulation, while 57% were oral anticoagulant-naïve. According to the timing of scheduled CV, two strategies have been investigated: an early strategy, in which the anticoagulant was given 1–5 days before CV (in patients randomized to rivaroxaban, medication was started at least 4 h before CV) and a delayed CV approach that intended to adequately anticoagulated patients for at least 3 weeks before the procedure. VKA anticoagulation was considered adequate if the INR was maintained in the range 2–3 for at least 3 consecutive weeks prior to CV. Prophylaxis with rivaroxaban was considered acceptable if the pill count was ≥80% in the 3 weeks preceding the CV. Fifty-eight percent of patients underwent early CV. A TEE was performed in 65% and 10% of patients treated with an early- or delayed cardioversion strategy, respectively. Almost all of CVs (97.6%) were electrical.

At 30 days of follow-up, the primary efficacy endpoint (a composite of stroke, transient ischemic attack, peripheral embolism, myocardial infarction, and cardiovascular death) occurred in 0.51% patients in the rivaroxaban arm and in 1.02% in the VKA arm (risk ratio of 0.50; 95% confidence interval (CI) 0.15–1.73). Major bleeding occurred in 0.6% patients in the rivaroxaban arm and in 0.8% patients in the VKA arm (risk ratio 0.76; 95% CI 0.21–2.67).

The results from X-VeRT trial suggest a trend towards a lower rate of thromboembolic events and major bleeding in the rivaroxaban group, in the whole population as well as in the early and delayed CV subgroups.

Furthermore, the average time to non-TEE-guided CV was shorter with rivaroxaban compared to VKA (median time 22 vs. 30 days, respectively; *p* < 0.001), suggesting that the use of rivaroxaban may reduce delay in cardioversion time [49].

### 6.2. Edoxaban

To compare edoxaban with warfarin in the peri-cardioversion setting, a post hoc analysis of ENGAGE AF-TIMI 48 (the Effective aNticoaGulation with factor Xa next GEneration in Atrial Fibrillation–Thrombolysis In Myocardial Infarction study 48) trial evaluated outcomes of 365 patients (632 ECV attempts) undergoing ECV on average 348 days post-randomization. No data has been collected about the rate of TEE-guided cardioversions performed during the trial.

Thirty days after CV, two thromboembolic events were observed in patients on the lower-dose edoxaban; none occurred in patients on warfarin or higher-dose edoxaban. There were no major bleeding events and one death in the high-dose edoxaban treatment. At 25 months of follow-up the trial has shown no statistically significant differences between treatment groups in terms of primary efficacy and safety endpoints.

To address the limitations of this retrospective post-hoc analysis, the ENSURE-AF (Edoxaban vs. enoxaparin/warfarin in patients undergoing cardioversion of atrial fibrillation) study was conducted. It randomized (1:1) 2199 patients scheduled for cardioversion to edoxaban 60 mg or enoxaparin/warfarin.

Patients were stratified based on cardioversion approach (TEE or non-TEE guided), the patient’s previous experience with OAC therapy at the time of randomisation (OAC-experienced or OAC-naive), edoxaban dose (60 mg daily or reduced 30 mg daily dose), and region. The dose of edoxaban was reduced to 30 mg in patients with one or more of the following clinical factors: moderate or severe renal impairment (creatinine clearance (CrCl) 15–50 mL/min), low body weight ≤ 60 kg, or concomitant use of certain P-glycoprotein (P-gp) inhibitors.

For patients who underwent TEE-guided cardioversion, cardioversion was performed within 3 days of randomization, after at least the first drug dose. In patients randomly assigned to the edoxaban, arm medication was started at least 2 h before ECV. In patients who had not received a TEE, therapeutic anticoagulation with warfarin for 3 weeks prior to cardioversion was required. The follow-up duration was 28 days on the study drug after CV plus another 30 days to assess safety.

The primary efficacy endpoint (composite of stroke, systemic embolism, myocardial infarction, or cardiovascular death) occurred in <1%) patients in the edoxaban arm vs. 1% in the enoxaparin–warfarin arm (odds ratio [OR] 0.46, 95% CI 0.12–1.43). The primary safety endpoint (composite of major and clinically relevant non major bleeding) occurred in 1% in the edoxaban arm vs. 1% in the enoxaparin–warfarin arm (OR 1.48, 95% CI 0.64–3.55).

No differences in outcome rates have been found regardless of the strategy of CV (TEE- vs. non-TEE-guided approach), prior anticoagulant use and the edoxaban dose adjustment [48].

Unlike X-VeRT trial, no significant difference between the two treatment groups was observed in the median time between randomization and cardioversion in both groups: (2 days in the TEE-guided group and 23 days in the non-TEE-guided group) [49].

This relevant difference between the two studies that have investigated rivaroxaban and edoxaban in the peri-cardioversion setting is likely to be due to the rigorous heparin protocol in the warfarin arm in the ENSURE-AF study [50]. However, it is important to keep in mind that this heparin/VKA protocol may not correspond to normal clinical practice.

### 6.3. Apixaban

A post hoc analysis of the ARISTOTLE trial evaluated the rate of stroke, systemic embolism, myocardial infarction, major bleeding, and death, in 540 patients naïve from any other OAC therapy who underwent CV during trial follow-up.

A pre-cardioversion TEE was performed in about one third of cases, a rate similar to that reported in patients assigned to the dabigatran group and who had undergone CV in the RE-LY trial.

In the 30 days after CV no thromboembolic events occurred; one myocardial infarction and one major bleeding occurred in each group; and two patients died in each of the apixaban and warfarin groups.

The EMANATE trial was the only trial conducted to compare a NOAC (apixaban) to heparin plus VKA in patients with NVAF who were anticoagulation-naïve, defined as having received less than 48 h of anticoagulation, whereas the most previous evidence of NOACs in the setting of CV concerns to anticoagulated patients with AF of 48 h duration or longer [51]. In the ENSURE-AF study only about 27% of all patients were naive to OAC, defined as not having received any OAC within 30 days before randomization, whereas in the X-VeRT trial about 57% were not oral anticoagulant experienced, defined as oral anticoagulant use <6 weeks to first study medication intake (approximately 23% of patients managed for newly diagnosed AF) [49,50].

The EMANATE trial included 1.500 patients randomized (1:1) to receive apixaban 5 mg twice daily (or dose reduced per standard criteria), with or without a single loading dose to expedite cardioversion, or standard care (heparin plus warfarin). The choice of an imaging-guide strategy (TEE or CT imaging) as the usage of the loading dose in patients randomized to apixaban were at the discretion of the investigator.

A pre-procedural imaging study was performed in 57% of cases. In patients randomized to apixaban if the imaging study showed no clot, five doses of apixaban were administered to achieve steady-state concentration before CV. Alternatively, if no clot was detected, an immediate CV could be undertaken following a single 10 mg loading dose of apixaban, or dose reduced per standard criteria, administered at least 2 h before CV, followed by a maintenance regimen. In the apixaban group, about half of patients were treated with an initial loading dose of 10 mg. For patients in the standard care arm, similar timing was permitted after administration of heparin.

Anticoagulation was administered from randomization until 30 days after CV, and for a maximum of 90 days if CV was not performed.

Compliance was estimated by pill count and in the apixaban arm; 91% of patients had a compliance >80%.

No patients in the apixaban arm had a stroke vs. six in the standard care arm (*p* = 0.015). Systemic embolic events did not occur in either arm. Two deaths occurred in the apixaban arm vs. one in the standard care arm. Bleeding rate was also similar, with three of the patients in the apixaban arm had major bleeding and 11 had clinically relevant non-major (CRNM) vs. six and 13 in the standard care arm (*p* = 0.33 and *p* = 0.68, respectively) [51].

Sixty-one patients had LAAT (about 7%), 30 in the apixaban arm and 31 in the standard care arm. Thrombus resolution percentage was similar in patients treated with apixaban (52%) as with standard care therapy (56%). In patients with LAAT, no adverse events occurred in both treatment arm.

## 7. Recommendations from ESC Guidelines and Consensus Documents for the Use of NOACs in Peri-Cardioversion Setting

Based on current ESC guidelines [4] in patients with NVAF undergoing CV, NOACs are recommended with at least similar efficacy and safety as VKA (Class IA) Figure 3. Anticoagulation with heparin or NOACs should be started as soon as possible before CV (Class IIA). Moreover, regarding the type of strategy to choose in OAC-naive patients with AF of ≥48 h (or unknown) duration, ESC guidelines [4] and the European Heart Rhythm Association (EHRA) consensus document [8] suggest two options: an early imaging-guided strategy or a delayed non-imaging-guided strategy after regular and continued NOAC intake for at least 3 weeks before CV.

For this clinical scenario, X-VeRT, ENSURE-AF, and EMANATE studies offered important data since most of the patients enrolled had an AF of ≥48 h duration (all patients in the first two studies).

When the early strategy is adopted a standard initial NOAC dose (Rivaroxaban 20/15 mg, Edoxaban 60/30 mg, Dabigatran 150/110 mg) must be administered >4 h before cardioversion (≥2 h after apixaban loading dose) and a TEE or a CT imaging must be performed to exclude LAAT [49,50,51]. According to EMANATE trial data an initial loading dose of 10 mg of apixaban (5 mg if does-adjustment criteria are applied) should be administered. Regarding the other NAC, a loading dose is not recommended [8].

If a LAAT is found, cardioversion must be postponed after a longer period of anticoagulation after a repeated imaging test to confirm thrombus resolution. In this setting the best therapeutic strategy is not established. There are several possibilities: converting to heparin/VKA or start or continue with NOACs (best data with rivaroxaban and apixaban) especially in patients on VKA with poor anticoagulation quality (low time in therapeutic range).

Limited data are instead available on efficacy and safety of NOACs for OAC-naïve patients with NVAF of <48 h duration who are usually cardioverted without TEE after a single dose of LMWH. The last EHRA consensus document on NOACs in NVAF recommends for these patients to follow the local protocol with heparin/VKA as their first choice. An alternative strategy may be the use of a single dose of NOACs or a loading dose of apixaban 2–4 h before CV to replace LMWH, with or without TEE, according to patient thromboembolic risk and AF duration [25,52].

However, none of the NOAC studies on the peri-cardioversion setting including the EMANATE trial (the only study to enroll a certain number of OAC-naive patients with AF of <48 h duration) demonstrated non-inferiority in terms of efficacy and safety of a single dose of NOACs or a single loading dose of apixaban compared to LMWH in this clinical scenario [51].

The duration of anticoagulation with NOACs as for VKA post-cardioversion depends on the individual patient’s thromboembolic risk assessed with CHA2DS2-VASc score. Men and women with a CHA2DS2-VASc ≥ 1 and ≥2, respectively, require long-term OAC therapy regardless of cardioversion success. For patients with AF duration > 48 h and CHA2DS2-VASc score 0 in men and 1 in women, OAC therapy needs to be continued for 4 weeks post-CV. In contrast, the optimal duration of anticoagulation in AF ≤ 48 h (especially when <12 h) is unknown. In conclusion, anticoagulation with NOACs appear effective and safe in the peri-cardioversion setting [53].

## 8. Controversial Issues

### 8.1. The Need of Image-Guided Strategy to Exclude Atrial Thrombi

Although outcomes associated with the image-guided strategy were not predefined endpoints in these studies, with the exception of the ENSURE-AF trial, it is interesting that similar outcomes were associated with conventional and TEE-guided strategies.

Cumulative data from these studies suggested that a non-TEE-guided approach is reasonably safe for patients who have been on a properly dosed NOAC with 3 weeks of therapy before CV. It is important to note that with NOACs routine laboratory tests to evaluate effective anticoagulation are not recommended, and therefore the patient needs to be carefully questioned about adherence during the 3 weeks before CV. If there is any doubt about adherence, TEE should be performed prior to cardioversion during NOAC treatment.

The decision to perform TEE before CV should be made on a case-by-case basis according to the individual CHA2DS2-VASc score. In fact, in 1.6–2.1% of anticoagulated patients TEE performed prior to AF ablation revealed LAAT or sludge in the left atrium, with the risk of thrombus correlating with the CHADS2 score (thrombus incidence ≤ 0.3% in patients with CHADS2 score 0–1 and 0.5% in patients with CHADS2 score ≥ 2, respectively) [54,55,56].

Moreover, an unexpected high rate of LAAT thrombi has been found in anticoagulated patients (with both NOACs or VKAs) in NOACs trial (3% in X-VeRT; 7% in EMANATE) [49,50,51].

### 8.2. NOACs Efficacy and Safety According to Thromboembolic Risk

The thromboembolic risk profile of patients enrolled in the four phase III trials on NVAF is largely heterogenous, with subgroups of patients undergoing CV presenting a lower risk than the whole population. The prospective trials, discussed above, also enrolled patients with different thromboembolic risk. In ENSURE-AF and EMANATE the mean CHA2DS2-VASc scores were 2.6 and 2.8, respectively. The proportion of patients that were moderate-to-high risk (score ≥ 2) was 76% and 100%, respectively, whereas the X-VERT population had a relatively lower risk (mean CHA2DS2-VASc score 2.3; approximately 64% with CHA2DS2-VASc score ≥ 2). The variable risk of such a population prevented us from conducting an evaluation of NOACs efficacy and safety according to thromboembolic risk [46,47,48].

### 8.3. Patients with Renal Dysfunction

Despite the promising results reported, we have little data on patients with renal failure who underwent cardioversion after NOAC treatment. Patients with severe renal failure were excluded from the post-hoc analysis on cardioversion of the four main landmark trials of NOACs use in NVAF.

In the X-VeRT trial a reduced dose of rivaroxaban 15 mg daily was used in patients with a CrCl between 30 and 49 mL/min while patients with a CrCl < 30 mL/min were excluded. However, only a few patients (6.8%) in the rivaroxaban arm had a CrCl ≤ 50 mL/min [49]. The ENSURE-AF study excluded patients with a CrCl <15 mL/min and a reduced dose of edoxaban 30 mg daily was used in patients with a CrCl between 15 and 50 mL/min but only 8% of patients in the edoxaban arm had a CrCl ≤ 50 mL/min [50]. In the EMANATE trial a reduced loading dose of 5 mg and a reduced maintenance dose of 2.5 mg of apixaban twice daily was used in patients with two of the following criteria: age ≥ 80 years, weight ≤ 60 kg, or serum creatinine ≥ 1.5 mg/dL, but only 11 patients received a reduced loading dose of 5 mg and no data are shown on the percentage of patients in the apixaban arm that underwent cardioversion without loading dose [51].

Prevention of thromboembolic events in patients with AF and severe CKD (chronic kidney disease) is a difficult task for physicians and an adequate anticoagulant strategy and regular renal function monitoring should be maintained for these patients. Stage 4 CKD seems to be independently associated with an increment of thrombin generation and a lower fibrinolysis capacity, regardless the stroke risk factors [57].

Patients with end-stage CKD (stage 5) have greater risk for AF, stroke/SE, and bleeding. Data from observational studies suggest improved safety and convenience in NOAC treated patients compared with VKA but there is no solid evidence for embolic events reduction with either NOAC or VKA. Notably, NOACs have not been approved in Europe for patients with CrCl ≤ 15 mL/min or on dialysis [4].

### 8.4. Patients Undergoing Pharmacological Cardioversion

Current guidelines do not indicate a preference for cardioversion modality (ECV or PCV) based on the type of anticoagulant treatment (NOACs vs. VKA). However, except for the use of rivaroxaban in the post hoc analysis of the ROCKET AF trial, almost all patients included in NOACs trials received ECV [46].

### 8.5. Early- vs. Delayed Strategy

All studies conducted so far have failed to demonstrate the superiority of an early- vs. delayed strategy in terms of sinus rhythm reestablishment and maintenance. No difference in the acute CV success rate was observed in the X-VeRT or ENSURE-AF trial between patients randomized to the early- or delayed strategy [49,50]. In the EMANATE trial, although the early strategy (loading dose of apixaban + imaging) resulted in a very short time from randomization to cardioversion, the success rate of cardioversion between the different groups did not differ [51].

However, in order to evaluate the early- vs. delayed strategy efficacy in maintaining sinus rhythm, data from larger RCTs with longer follow-up are needed.

## 9. Conclusions

Cardioversion is a well-recognized procedure which is part and parcel of a rhythm control strategy in AF patients. However, patients with recent-onset AF could undergo to a wait-and-watch approach, since there are many chances to convert spontaneously within 48 h. The decision to adopt the rhythm control approach should be based on a specific pattern which may help to forecast AF recurrence.

Cardioversion should be performed after a careful evaluation of thromboembolic risk before the procedure, starting timely OAC, and continuing it according to stroke risk. The NOACs has brought about changes in peri-procedural anticoagulation management allowing performance of cardioversion without major delays, provided that patients are adequately compliant with NOAC treatment. After the procedure, a punctual clinical follow-up is needed lest AF might recur.

## Figures and Tables

**Figure 1 jcm-10-03212-f001:**
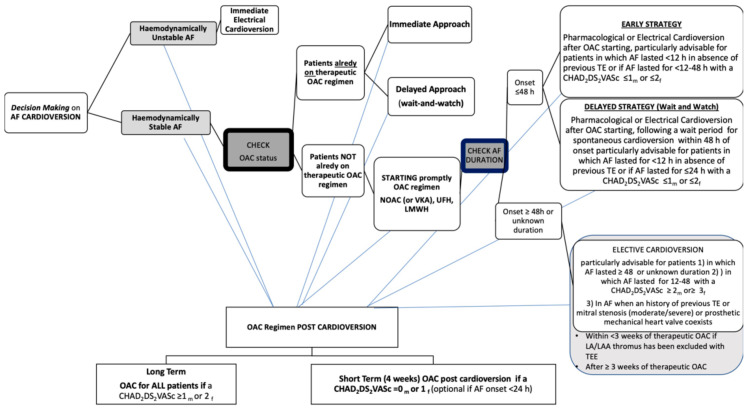
Decision making on atrial fibrillation cardioversion according to 2020 ESC Guidelines. AF, atrial fibrillation; OAC, oral anticoagulant; NOAC, non-vitamin K antagonist oral anticoagulant;VKA, vitamin K antagonist; UFH, unfractionated heparin; LMWH, low-molecular-weight heparin; TE, thromboembolism; h, hour; CHA2DS2-VASc, congestive heart failure, hypertension, age ≥ 75 years, diabetes mellitus, stroke, vascular disease, age 65–74 years, sex category (female); m, male; f, female; LA, left atrium; LAA, left atrial appendage; TEE, transesophageal echocardiography.

**Figure 2 jcm-10-03212-f002:**
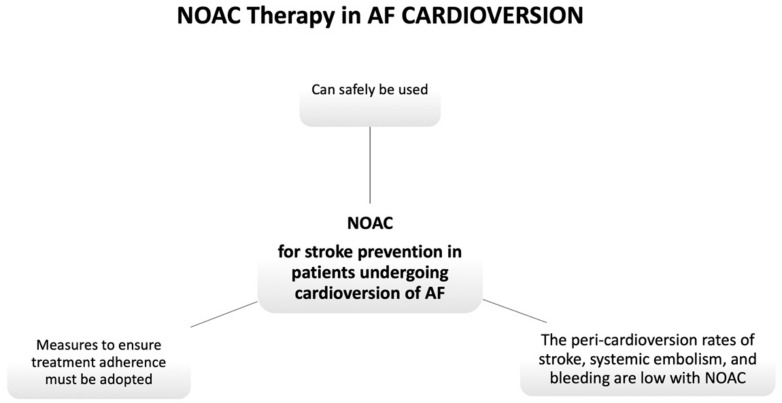
NOAC Therapy in AF CARDIOVERSION. NOAC, non-vitamin K antagonist oral anticoagulant.

**Figure 3 jcm-10-03212-f003:**
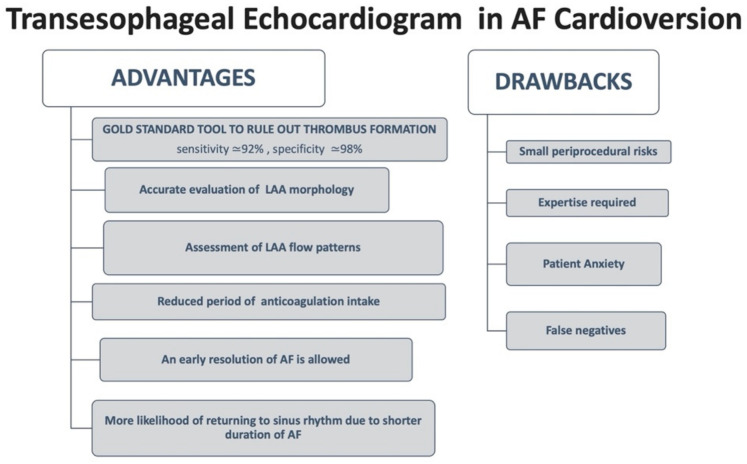
Transesophageal Echocardiogram in AF Cardioversion. LAA, left atrial appendage; AF, atrial fibrillation.

**Table 1 jcm-10-03212-t001:** Post hoc analyses of NOAC randomized clinical trials on anticoagulation in patients undergoing cardioversion.

	RELY [37]	ROCKET-AF [38]	ARISTOTLE [39]	ENGAGE AF TIMI [40]
NOAC	Dabigatran	Rivaroxaban	Apixaban	Edoxaban
Patients	1270	285	540	365
CV	1983	375	743	632
Comparator	Warfarin	Warfarin	Warfarin	Warfarin
TEE-guided CV	21%	NA	27%	NA
Follow-up	30 days	30 days	30 days	30 days
Stroke Or Systemic Embolism	11 (0.6%)	2 (0.7%)	0 (0%)	0 (0%)
NOAC vs. VKA	7 (0.5%) vs. 4 (0.6%)	NA	-	-
Major bleeding	19 (1.0%)	NA	2 (0.2%)	0 (0%)
NOAC vs. VKA	15 (1.1%) vs. 4 (0.6%)	NA	1 (0.3%) vs. 1 (0.2%)	-

CV, cardioversion; NA, non available; NOAC, non-vitamin K oral anticoagulant; TEE, transesophageal echocardiography; VKA, vitamin K-antagonist.

**Table 2 jcm-10-03212-t002:** Randomized trials on NOACs in the setting of cardioversion.

	X-VeRT [41]	ENSURE AF [42]	EMANATE [43]
NOAC	Rivaroxaban	Edoxaban	Apixaban
Study design	Open-label, randomized 2:1	Open-label, randomized 1:1	Open-label, randomized 1:1
Patients	1504	2199	1500
AF duration	≥48 h	≥48 h	<48 h and ≥48 h
Comparator	VKA	LMWH/VKA	Heparin/VKA
Treatment strategy	TEE vs. no TEE	TEE vs. no TEE	Imaging vs. no Imaging
TEE-guided CV	Riva 67%, VKA 65%	100%	100%
Early strategy	Adeguate anticoagulant or TEE + Rivarovaban at least 4 h before CV	Adeguate anticoagulant or TEE+ Edoxaban at least 2 h before CV	Imaging + Loading dose Apixaban at least 2 h before CV or after five doses
Post-procedural anticoagulation	42 days	28 days	30 days
Primary efficacy endpoint	Composite of stroke, TIA, SE, MI and CV death	Composite of stroke, TIA, SE, MI and CV death	Stroke, SE, Death
NOAC vs. VKA	5 (0.5%) vs. 5 (1.0%)	5 (0.5%) vs. 11 (1.0%)	0 (0%) vs. 6 (0.8%)
Primary safety end point	ISTH Major bleeding	ISTH Major or CRNM bleeding	ISTH Major or CRNM bleeding
NOAC vs. VKA	6 (0.6%) vs. 4 (0.8%)	16 (1.5%) vs. 11 (1.0%)	14 (1.9%) vs. 19 (2.5%)
Difference in time to CV	Yes	No	No

AF, atrial fibrillation; CRNM, clinically relevant non-major; CV, cardioversion; h, hours; LMWH, low molecular weight heparin; ISTH, International Society of Thrombosis and Haemostasis; NOAC, non-vitamin K oral anticoagulant; SE, systemic embolism; TIA, transitory ischemic attack; TEE, transesophageal echocardiography; VKA, vitamin K-antagonist.

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
