# Peer review of "Anticoagulation in Atrial Fibrillation Cardioversion: What Is Crucial to Take into Account"

_jcm, 2021, doi:10.3390/jcm10153212_

Round 1
Reviewer 1 Report
Dear Editors and Authors,
I read article entitled 'Anticoagulation in Atrial Fibrillation Cardioversion: What Is Crucial to Take Into Account' with interest.
This review article concerns important topic in cardiovascular medicine: anticoagulation in atrial fibrillation (AF). The general structure of the paper is accurate.
However, I have some comments to further improve the paper:
- What should be the definition of hemodynamic instability in the case of AF qualifying for emergency ECV?
- The authors state: “Moreover, maximum fixed-energy electrical CV was more effective than an energy-escalation strategy.” Please provide details on recommended energy (or which factors should be taken into account while calculating this parameter) for ECV with appropriate references.
- Please be more specific also in the sentences “spontaneous echo contrast and decreased atrial emptying velocities” and “Likewise, the presence of normal velocities in appendage seems to be associated with a successful ECV” – which velocities should be considered as decreased/normal?
- Moreover, in the sentence “When the early strategy is adopted a single NOAC dose must be administered > 4h before cardioversion (≥ 2h after apixaban loading dose)” please precise the dosage of all NOACs before cardioversion.
- The decision on rhythm control strategy is very important in taking care of patients with AF. Please provide LAVI, left atrial diameter and atrial conduction delays thresholds (with appropriate references) suggesting the need for cardioversion/rhythm control strategy.
- Should the kidney function be assessed when taking into account decision on introduction of anticoagulation (please see e.g.: Clin. Med.2020, 9(8), 2476; https://doi.org/10.3390/jcm9082476)?
- Please provide the figure with your suggestions on anticoagulation – based on/similar to Figure 16 in the most recent ESC guidelines. In which places the authors opinions differ from the opinions of the ESC guidelines writing committee? Or could the authors provide this assessment as the algorithm for practicing clinicians?
- The authors state in the abstract: “In patients with AF occurring within less than 48 hours, synchronized direct ECV should be the elective procedure, as it restores sinus rhythm quicker and more successfully than pharmacological cardioversion (PCV) and is associated with shorter length of hospitalization.”, while in the introduction: “whereas pharmacological cardioversion (PCV) may be preferred in recent-onset AF4” these statements are contradictory.
- What do the authors mean by writing “Due to the prolonged atrial conduction” regarding WPW syndrome?
- The authors state: “AF itself independently increases stroke risk by 5-fold, which represents one of the leading causes of morbidity and death in AF patients”. How about comorbidities/risk factors – are there clear data showing that AF itself increases this risk to so high extent? Please see e.g.: DOI: https://doi.org/10.1016/j.cjca.2019.01.014?
- Please write conclusions in the main part of the paper.
- Table 1. Why line under verse 1 is incomplete? Another 2 lines should separate the verses on Stroke Or Systemic Embolism + NOAC vs. VKA and Major bleeding + NOAC vs. VKA. All abbreviations should be explained below the table. Please change “na” into “NA”.
- Typos/grammatic errors: “renal disfunction” or “questionnaire maintains the same safety than a TOE-guided approach”.
- Reference number 49 is incomplete.
Author Response
(Reviewer 1)
- What should be the definition of hemodynamic instability in the case of AF qualifying for emergency ECV?
- We agree with the reviewer. We provided the definition:
Acute haemodynamic instability in AF, due to a rapid ventricular rate (typically >150 bpm or higher in patients compromised by co-morbidities), is characterized by clinical manifestations such as syncope, acute pulmonary oedema, myocardial ischemia, symptomatic hypotension, or cardiogenic shock. In those patients an emergency electrical cardioversion has to be promptly performed, and anticoagulation should be started as soon as possible.4
Reference 4. Hindricks G, Potpara T, Dagres N, et al. 2020 ESC Guidelines for the diagnosis and management of atrial fibrillation developed in collaboration with the European Association for Cardio-Thoracic Surgery (EACTS) Eur Heart J. 2021;42(5):373-498
- The authors state: “Moreover, maximum fixed-energy electrical CV was more effective than an energy-escalation strategy.” Please provide details on recommended energy (or which factors should be taken into account while calculating this parameter) for ECV with appropriate references.
- We agree with the reviewer . We provided details on recommended energy with the appropriate reference: It has been claimed that an initial synchronised shock at maximum defibrillator output ( 360 J ) was a reasonable approach without an increasing in adverse events.14What is particularly noticeable is that an initial energy setting of  360 J resulted more efficiently than traditional approach, particularly when the duration of AF is longer15.
Reference 15. Hamada T, Hiraki T, Ikeda H, et al. Mechanisms for Atrial Fibrillation in Patients with Wolff-Parkinson-White Syndrome. J Cardiovasc Electrophysiol 2002;13(3):223–9.
- Please be more specific also in the sentences “spontaneous echo contrast and decreased atrial emptying velocities” and “Likewise, the presence of normal velocities in appendage seems to be associated with a successful ECV” – which velocities should be considered as decreased/normal?
-We agree with the reviewer, so we clarified that ‘‘smoke effect described as a swirling pattern of increased echogenicity, and decreased atrial emptying velocities (peak LAA velocity<20 cm/s), are associated with a greater risk of stroke or peripheral embolism. It is important to recognize that also LA dilation, reduced LA strain, LAA thrombus, and non-chicken wing LAA morphology have been proposed as predictive factors of stroke.38Likewise, the presence of normal velocities in appendage (n.v. 20–40 cm/s) 39seems to be associated with a successful ECV, and with the maintenance of long-term sinus rhythm40, and as it turned out, a mean LAA peak emptying velocity peak  >40 cm/s has proven proved to be independent predictors of one-year no AF recurrence40.
References:
- Delgado V, Di Biase L, Leung M, Romero J, Tops LF, Casadei B, Marrouche N, Bax JJ. Structure and function of the left atrium and left atrial appendage: AF and stroke implications. J Am Coll Cardiol 2017;70:3157-3172
- Pollick C., Taylor D. Assessment of left atrial appendage function by transesophageal echocardiography. Implications for the development of thrombus. Circulation. 1991;84:223–231
- Antonielli E, Pizzuti A, Pálinkás A, et al. Clinical value of left atrial appendage flow for prediction of long-term sinus rhythm maintenance in patients with nonvalvular atrial fibrillation. J Am Coll Cardiol 2002;39(9):1443–9.
- Moreover, in the sentence “When the early strategy is adopted a single NOAC dose must be administered > 4h before cardioversion (≥ 2h after apixaban loading dose)” please precise the dosage of all NOACs before cardioversion.
-We agree with the reviewer, so it has been discussed that “When the early strategy is adopted a standard initial NOAC dose (Rivaroxaban 20/15 mg, Edoxaban 60/30 mg, Dabigatran 150/110 mg) must be administered > 4h before cardioversion (≥ 2h after apixaban loading dose) and a TEE or a CT imaging must be performed to exclude LAAT49-51. According to EMANATE trial data an initial loading dose of 10 mg of apixaban (5 mg if does-adjustment criteria are applied) should be administered. Regarding as the other NAC, a loading dose it has not been advised8.
References: 49. Cappato R, Ezekowitz MD, Klein AL, et al; X-VeRT Investigators . Rivaroxaban vs. vitamin K antagonists for cardioversion in atrial fibrillation. Eur Heart J 2014;35(47):3346–55.
- Goette A, Merino JL, Ezekowitz MD, et al. Edoxaban versus enoxaparin/warfarin in patients undergoing cardioversion of atrial fibrillation (ENSURE-AF): a randomised, open-label, phase 3b trial. Lancet 2016;388(10055):1995–2003.
- Ezekowitz MD, Pollack Jr C V, et al. Apixaban compared to heparin/vitamin K antagonist in patients with atrial fibrillation scheduled for cardioversion: the EMANATE trial. Eur Heart J.2018;39(32):2959–71.
- The decision on rhythm control strategy is very important in taking care of patients with AF. Please provide LAVI, left atrial diameter and atrial conduction delays thresholds (with appropriate references) suggesting the need for cardioversion/rhythm control strategy.
-We agree with the reviewer, so it has been discussed that “Factors which might facilitate an attempt at rhythm control should be evaluated4.
It has been claimed that that left atrial volume index (LAVI) could represent an effective marker in predicting the maintenance of SR contributing to identify patients in which CV is likely to be successful. A cut-off 55 mL/m2 has been proposed.32Another factor which should be be taken into account is left atrial diameter since it seems to be associated with AF recurrence after cardioversion if it is larger than 44 mm33. Conversely it could be uttered that an heterogeneous electrical activation of the LA appears to be related to AF recurrence. An advanced interatrial block (aIAB), P wave duration >120 ms and biphasic P waves in the inferior leads have been recognized as independent predictors of AF recurrence.34
References: 32. Toufan M, Kazemi B, Molazadeh N. The significance of the left atrial volume index in prediction of atrial fibrillation recurrence after electrical cardioversion. J Cardiovasc Thorac Res. 2017; 9(1): 54–59.
- Efremidis M, Alexanian IP, Oikonomou D, et al. Predictors of atrial fibrillation recurrence in patients with long-lasting atrial fibrillation. Can J Cardiol 2009;25(4):e119-e124
- Baranchuk A, Yeung C. Advanced interatrial block predicts atrial fibrillation recurrence across different populations: learning Bayes syndrome. Int J Cardiol 2018;272:221-222.
- Should the kidney function be assessed when taking into account decision on introduction of anticoagulation (please see e.g.: Clin. Med.2020, 9(8), 2476; https://doi.org/10.3390/jcm9082476)?
-We agree with the reviewer, so it has been said that “there can be no doubt that prevention of thromboembolic events in patients with AF and severe CKD (Chronic kidney disease) looks for physicians an extremely sticky wicket so that an adequate anticoagulant strategy and regular renal function monitoring should be carefully kept in mind in these patients. Stage 4 CKD seems to be independently associated with an increment of thrombin generation and a lower fibrinolysis capacity, regardless the stroke risk factors .57
Patients with end-stage CKD (stage 5) have greater risk for AF, stroke/SE, and bleeding. Data from observational studies suggest improved safety and convenience in NOAC treated patients compared with VKA but there is no solid evidence for embolic events reduction with either NOAC or VKA. Notably, NOAC have not been approved in Europe for patients with CrCl <_15 mL/min or on dialysis4.”
Reference 57. Matusik PT , Heleniak Z, Papuga-Szela E , Plens K, Lelakowski J , Undas A. Chronic Kidney Disease and Its Impact on a Prothrombotic State in Patients with Atrial Fibrillation. Clin Med. 2020 1;9(8):2476.
- Please provide the figure with your suggestions on anticoagulation – based on/similar to Figure 16 in the most recent ESC guidelines.
-We agree with the reviewer; there can be no doubt that a figure similar to the one in the ESC guidelines makes a good impression, so we provided a flowchart (fig.1)
- In which places the authors opinions differ from the opinions of the ESC guidelines writing committee? Or could the authors provide this assessment as the algorithm for practicing clinicians?
-We agree with the reviewer, so we suggested to comply with ESC guidelines as it has been summarized in the figure: “As far as we are concerned, the most recent ESC guidelines should be considered an useful guide in order to avoid fraught roads with danger and tricky situations as has been summarized in figure 1.”
- The authors state in the abstract: “In patients with AF occurring within less than 48 hours, synchronized direct ECV should be the elective procedure, as it restores sinus rhythm quicker and more successfully than pharmacological cardioversion (PCV) and is associated with shorter length of hospitalization.”, while in the introduction: “whereas pharmacological cardioversion (PCV) may be preferred in recent-onset AF4” these statements are contradictory.
We agree with the reviewer, the phrase “whereas pharmacological cardioversion (PCV) may be preferred in recent-onset AF” has been deleted. On the other hand it was said that “in stable patients, pharmacological and electrical cardioversion can be performed both. Electrical cardioversion is more effective but, in contrast, sedation is needed”.
- What do the authors mean by writing “Due to the prolonged atrial conduction” regarding WPW syndrome?
-We agree with the reviewer and we clarified the sentence: “Due to the prolonged atrial conduction WPW, characterized by a longer maximal atrial conduction delay and wider conduction delay zone, the atrial vulnerability to develop of AF is greater16.
- Reference 16 Hamada T, Hiraki T, Ikeda H, et al. Mechanisms for Atrial Fibrillation in Patients with Wolff-Parkinson-White Syndrome. J Cardiovasc Electrophysiol 2002;13(3):223–9.
- The authors state: “AF itself independently increases stroke risk by 5-fold, which represents one of the leading causes of morbidity and death in AF patients”. How about comorbidities/risk factors – are there clear data showing that AF itself increases this risk to so high extent? Please see e.g.: DOI: https://doi.org/10.1016/j.cjca.2019.01.014?
- We agree with the reviewer and we debated this issue: “The annual risk of stroke in those patients is greater than 20%. It is important to recognize that in absence of additional clinical stroke risk, assessed by CHA2DS2-VASc score, AF patients without anticoagulant therapy have an ischemic stroke rate of 0.43% per year. Conversely in those with 1 additional point in the CHA2DS2-VASc score the risk range from 1.18% to 3.50% per year. On the other hand, another commonly held claim is that an increased risk of stroke and systemic thromboembolism in AF is, in a certain way, linked to a persistent prothrombotic state, as demonstrated by the increasing of platelet activation, thrombin formation and inflammation with a reduction fibrinolysis process and by the endothelial dysfunction both. Unsurprisingly, what is particularly noticeable is that even young and very low-risk patients with AF have prothrombotic alterations, and the so-called prothrombotic fibrin clot phenotype could be often present. GÅ‚owicki and coworkers showed that a prothrombotic pathway might involve in increasing risk among patients with AF with the CHA2DS2-VASc score of 1 despite the sex.21”
- Reference 21. Głowicki B, Matusik PT, Plens K, Undas A. Prothrombotic State in Atrial Fibrillation Patients With One Additional Risk Factor of the CHA2DS2VASc Score (Beyond Sex). Can J Cardiol. 2019;35(5):634-643.
- Please write conclusions in the main part of the paper.We agree with the reviewer and we wrote conclusion
Conclusions
Cardioversion is a well recognized procedure which is part and parcel of a rhythm control strategy in AF patients. However, patients with recent-onset AF could undergo to a wait-and- watch approach, since there are many chances to convert spontaneously within 48 h. The decision to adopt the rhythm control approach should be based on a specific pattern which may help to forecast AF recurrence.
Cardioversion should be performed after a careful evaluation of thromboembolic risk before the procedure, starting timely OAC and continuing it life-long according to stroke risk. The NOACs has ring the changes on the peri-procedural anticoagulation management allowing to perform cardioversion without major delays, provided that patients have an adequate compliance to NOAC treatment. After the procedure, a punctual clinical follow-up is needed lest AF should recurre.
- Table 1. Why line under verse 1 is incomplete? Another 2 lines should separate the verses on Stroke Or Systemic Embolism + NOAC vs. VKA and Major bleeding + NOAC vs. VKA. All abbreviations should be explained below the table. Please change “na” into “NA”. Typos/grammatic errors: “renal disfunction” or “questionnaire maintains the same safety than a TOE-guided approach”.Reference number 49 is incomplete.
-We agree with the reviewer and we corrected the errors and we explaneid the abbreviations.
Figure 1 Decision making on Atrial Fibrillation cardioversion according to 2020 ESC Guidelines
AF,atrial fibrillation; OAC, oral anticoagulant; NOAC, non-vitamin K antagonist oral anticoagulant;VKA, vitamin K antagonist; UFH, unfractionated heparin; LMWH, low-molecular-weight heparin; TE, thromboembolism; h, hour; CHA2DS2-VASc, Congestive heart failure, Hypertension, Age >_75 years, Diabetes mellitus, Stroke, Vascular disease, Age 65 - 74 years, Sex category (female); m, men; f female; LA, left atrium; LAA, left atrial appendage; TEE, transoesophageal echocardiography.
(Reviewer 1)
- What should be the definition of hemodynamic instability in the case of AF qualifying for emergency ECV?
- We agree with the reviewer. We provided the definition:
Acute haemodynamic instability in AF, due to a rapid ventricular rate (typically >150 bpm or higher in patients compromised by co-morbidities), is characterized by clinical manifestations such as syncope, acute pulmonary oedema, myocardial ischemia, symptomatic hypotension, or cardiogenic shock. In those patients an emergency electrical cardioversion has to be promptly performed, and anticoagulation should be started as soon as possible.4
Reference 4. Hindricks G, Potpara T, Dagres N, et al. 2020 ESC Guidelines for the diagnosis and management of atrial fibrillation developed in collaboration with the European Association for Cardio-Thoracic Surgery (EACTS) Eur Heart J. 2021;42(5):373-498
- The authors state: “Moreover, maximum fixed-energy electrical CV was more effective than an energy-escalation strategy.” Please provide details on recommended energy (or which factors should be taken into account while calculating this parameter) for ECV with appropriate references.
- We agree with the reviewer . We provided details on recommended energy with the appropriate reference: It has been claimed that an initial synchronised shock at maximum defibrillator output ( 360 J ) was a reasonable approach without an increasing in adverse events.14What is particularly noticeable is that an initial energy setting of  360 J resulted more efficiently than traditional approach, particularly when the duration of AF is longer15.
Reference 15. Hamada T, Hiraki T, Ikeda H, et al. Mechanisms for Atrial Fibrillation in Patients with Wolff-Parkinson-White Syndrome. J Cardiovasc Electrophysiol 2002;13(3):223–9.
- Please be more specific also in the sentences “spontaneous echo contrast and decreased atrial emptying velocities” and “Likewise, the presence of normal velocities in appendage seems to be associated with a successful ECV” – which velocities should be considered as decreased/normal?
-We agree with the reviewer, so we clarified that ‘‘smoke effect described as a swirling pattern of increased echogenicity, and decreased atrial emptying velocities (peak LAA velocity<20 cm/s), are associated with a greater risk of stroke or peripheral embolism. It is important to recognize that also LA dilation, reduced LA strain, LAA thrombus, and non-chicken wing LAA morphology have been proposed as predictive factors of stroke.38Likewise, the presence of normal velocities in appendage (n.v. 20–40 cm/s) 39seems to be associated with a successful ECV, and with the maintenance of long-term sinus rhythm40, and as it turned out, a mean LAA peak emptying velocity peak  >40 cm/s has proven proved to be independent predictors of one-year no AF recurrence40.
References:
- Delgado V, Di Biase L, Leung M, Romero J, Tops LF, Casadei B, Marrouche N, Bax JJ. Structure and function of the left atrium and left atrial appendage: AF and stroke implications. J Am Coll Cardiol 2017;70:3157-3172
- Pollick C., Taylor D. Assessment of left atrial appendage function by transesophageal echocardiography. Implications for the development of thrombus. Circulation. 1991;84:223–231
- Antonielli E, Pizzuti A, Pálinkás A, et al. Clinical value of left atrial appendage flow for prediction of long-term sinus rhythm maintenance in patients with nonvalvular atrial fibrillation. J Am Coll Cardiol 2002;39(9):1443–9.
- Moreover, in the sentence “When the early strategy is adopted a single NOAC dose must be administered > 4h before cardioversion (≥ 2h after apixaban loading dose)” please precise the dosage of all NOACs before cardioversion.
-We agree with the reviewer, so it has been discussed that “When the early strategy is adopted a standard initial NOAC dose (Rivaroxaban 20/15 mg, Edoxaban 60/30 mg, Dabigatran 150/110 mg) must be administered > 4h before cardioversion (≥ 2h after apixaban loading dose) and a TEE or a CT imaging must be performed to exclude LAAT49-51. According to EMANATE trial data an initial loading dose of 10 mg of apixaban (5 mg if does-adjustment criteria are applied) should be administered. Regarding as the other NAC, a loading dose it has not been advised8.
References: 49. Cappato R, Ezekowitz MD, Klein AL, et al; X-VeRT Investigators . Rivaroxaban vs. vitamin K antagonists for cardioversion in atrial fibrillation. Eur Heart J 2014;35(47):3346–55.
- Goette A, Merino JL, Ezekowitz MD, et al. Edoxaban versus enoxaparin/warfarin in patients undergoing cardioversion of atrial fibrillation (ENSURE-AF): a randomised, open-label, phase 3b trial. Lancet 2016;388(10055):1995–2003.
- Ezekowitz MD, Pollack Jr C V, et al. Apixaban compared to heparin/vitamin K antagonist in patients with atrial fibrillation scheduled for cardioversion: the EMANATE trial. Eur Heart J.2018;39(32):2959–71.
- The decision on rhythm control strategy is very important in taking care of patients with AF. Please provide LAVI, left atrial diameter and atrial conduction delays thresholds (with appropriate references) suggesting the need for cardioversion/rhythm control strategy.
-We agree with the reviewer, so it has been discussed that “Factors which might facilitate an attempt at rhythm control should be evaluated4.
It has been claimed that that left atrial volume index (LAVI) could represent an effective marker in predicting the maintenance of SR contributing to identify patients in which CV is likely to be successful. A cut-off 55 mL/m2 has been proposed.32Another factor which should be be taken into account is left atrial diameter since it seems to be associated with AF recurrence after cardioversion if it is larger than 44 mm33. Conversely it could be uttered that an heterogeneous electrical activation of the LA appears to be related to AF recurrence. An advanced interatrial block (aIAB), P wave duration >120 ms and biphasic P waves in the inferior leads have been recognized as independent predictors of AF recurrence.34
References: 32. Toufan M, Kazemi B, Molazadeh N. The significance of the left atrial volume index in prediction of atrial fibrillation recurrence after electrical cardioversion. J Cardiovasc Thorac Res. 2017; 9(1): 54–59.
- Efremidis M, Alexanian IP, Oikonomou D, et al. Predictors of atrial fibrillation recurrence in patients with long-lasting atrial fibrillation. Can J Cardiol 2009;25(4):e119-e124
- Baranchuk A, Yeung C. Advanced interatrial block predicts atrial fibrillation recurrence across different populations: learning Bayes syndrome. Int J Cardiol 2018;272:221-222.
- Should the kidney function be assessed when taking into account decision on introduction of anticoagulation (please see e.g.: Clin. Med.2020, 9(8), 2476; https://doi.org/10.3390/jcm9082476)?
-We agree with the reviewer, so it has been said that “there can be no doubt that prevention of thromboembolic events in patients with AF and severe CKD (Chronic kidney disease) looks for physicians an extremely sticky wicket so that an adequate anticoagulant strategy and regular renal function monitoring should be carefully kept in mind in these patients. Stage 4 CKD seems to be independently associated with an increment of thrombin generation and a lower fibrinolysis capacity, regardless the stroke risk factors .57
Patients with end-stage CKD (stage 5) have greater risk for AF, stroke/SE, and bleeding. Data from observational studies suggest improved safety and convenience in NOAC treated patients compared with VKA but there is no solid evidence for embolic events reduction with either NOAC or VKA. Notably, NOAC have not been approved in Europe for patients with CrCl <_15 mL/min or on dialysis4.”
Reference 57. Matusik PT , Heleniak Z, Papuga-Szela E , Plens K, Lelakowski J , Undas A. Chronic Kidney Disease and Its Impact on a Prothrombotic State in Patients with Atrial Fibrillation. Clin Med. 2020 1;9(8):2476.
- Please provide the figure with your suggestions on anticoagulation – based on/similar to Figure 16 in the most recent ESC guidelines.
-We agree with the reviewer; there can be no doubt that a figure similar to the one in the ESC guidelines makes a good impression, so we provided a flowchart (fig.1)
- In which places the authors opinions differ from the opinions of the ESC guidelines writing committee? Or could the authors provide this assessment as the algorithm for practicing clinicians?
-We agree with the reviewer, so we suggested to comply with ESC guidelines as it has been summarized in the figure: “As far as we are concerned, the most recent ESC guidelines should be considered an useful guide in order to avoid fraught roads with danger and tricky situations as has been summarized in figure 1.”
- The authors state in the abstract: “In patients with AF occurring within less than 48 hours, synchronized direct ECV should be the elective procedure, as it restores sinus rhythm quicker and more successfully than pharmacological cardioversion (PCV) and is associated with shorter length of hospitalization.”, while in the introduction: “whereas pharmacological cardioversion (PCV) may be preferred in recent-onset AF4” these statements are contradictory.
We agree with the reviewer, the phrase “whereas pharmacological cardioversion (PCV) may be preferred in recent-onset AF” has been deleted. On the other hand it was said that “in stable patients, pharmacological and electrical cardioversion can be performed both. Electrical cardioversion is more effective but, in contrast, sedation is needed”.
- What do the authors mean by writing “Due to the prolonged atrial conduction” regarding WPW syndrome?
-We agree with the reviewer and we clarified the sentence: “Due to the prolonged atrial conduction WPW, characterized by a longer maximal atrial conduction delay and wider conduction delay zone, the atrial vulnerability to develop of AF is greater16.
- Reference 16 Hamada T, Hiraki T, Ikeda H, et al. Mechanisms for Atrial Fibrillation in Patients with Wolff-Parkinson-White Syndrome. J Cardiovasc Electrophysiol 2002;13(3):223–9.
- The authors state: “AF itself independently increases stroke risk by 5-fold, which represents one of the leading causes of morbidity and death in AF patients”. How about comorbidities/risk factors – are there clear data showing that AF itself increases this risk to so high extent? Please see e.g.: DOI: https://doi.org/10.1016/j.cjca.2019.01.014?
- We agree with the reviewer and we debated this issue: “The annual risk of stroke in those patients is greater than 20%. It is important to recognize that in absence of additional clinical stroke risk, assessed by CHA2DS2-VASc score, AF patients without anticoagulant therapy have an ischemic stroke rate of 0.43% per year. Conversely in those with 1 additional point in the CHA2DS2-VASc score the risk range from 1.18% to 3.50% per year. On the other hand, another commonly held claim is that an increased risk of stroke and systemic thromboembolism in AF is, in a certain way, linked to a persistent prothrombotic state, as demonstrated by the increasing of platelet activation, thrombin formation and inflammation with a reduction fibrinolysis process and by the endothelial dysfunction both. Unsurprisingly, what is particularly noticeable is that even young and very low-risk patients with AF have prothrombotic alterations, and the so-called prothrombotic fibrin clot phenotype could be often present. GÅ‚owicki and coworkers showed that a prothrombotic pathway might involve in increasing risk among patients with AF with the CHA2DS2-VASc score of 1 despite the sex.21”
- Reference 21. Głowicki B, Matusik PT, Plens K, Undas A. Prothrombotic State in Atrial Fibrillation Patients With One Additional Risk Factor of the CHA2DS2VASc Score (Beyond Sex). Can J Cardiol. 2019;35(5):634-643.
- Please write conclusions in the main part of the paper.We agree with the reviewer and we wrote conclusion
Conclusions
Cardioversion is a well recognized procedure which is part and parcel of a rhythm control strategy in AF patients. However, patients with recent-onset AF could undergo to a wait-and- watch approach, since there are many chances to convert spontaneously within 48 h. The decision to adopt the rhythm control approach should be based on a specific pattern which may help to forecast AF recurrence.
Cardioversion should be performed after a careful evaluation of thromboembolic risk before the procedure, starting timely OAC and continuing it life-long according to stroke risk. The NOACs has ring the changes on the peri-procedural anticoagulation management allowing to perform cardioversion without major delays, provided that patients have an adequate compliance to NOAC treatment. After the procedure, a punctual clinical follow-up is needed lest AF should recurre.
- Table 1. Why line under verse 1 is incomplete? Another 2 lines should separate the verses on Stroke Or Systemic Embolism + NOAC vs. VKA and Major bleeding + NOAC vs. VKA. All abbreviations should be explained below the table. Please change “na” into “NA”. Typos/grammatic errors: “renal disfunction” or “questionnaire maintains the same safety than a TOE-guided approach”.Reference number 49 is incomplete.
-We agree with the reviewer and we corrected the errors and we explaneid the abbreviations.
Figure 1 Decision making on Atrial Fibrillation cardioversion according to 2020 ESC Guidelines
AF,atrial fibrillation; OAC, oral anticoagulant; NOAC, non-vitamin K antagonist oral anticoagulant;VKA, vitamin K antagonist; UFH, unfractionated heparin; LMWH, low-molecular-weight heparin; TE, thromboembolism; h, hour; CHA2DS2-VASc, Congestive heart failure, Hypertension, Age >_75 years, Diabetes mellitus, Stroke, Vascular disease, Age 65 - 74 years, Sex category (female); m, men; f female; LA, left atrium; LAA, left atrial appendage; TEE, transoesophageal echocardiography.

Reviewer 2 Report
The article by Fabiana Luca et al. reviewed the latest trend and consensus for ‘the anticoagulation prior to and after cardioversion in patients with atrial fibrillation (AF)’, and also described the recent medical issues of anticoagulation in electrical cardioversion for AF.
This review article is well organized and especially describes the use of non-vitamin K-dependent oral anticoagulants (NOACs) in the setting of cardioversion.
In addition, the authors provide clinical evidence based on a lot of literature, and the review points are of scientific importance.
I have several minor comments:
- The authors used the ‘TOE’ as an abbreviation for transesophageal echocardiography. However, I think that the authors would be better to use ‘TEE’ rather than TOE because the general medical readers might not be familiar with this abbreviation of ‘TOE’.
- Page 2, line 90, “Electrical cardioversion in an emergency” : Please provide the anticoagulation strategy after urgent cardioversion for AF
- Please improve the quality of Table 1.
- Finally, the use of appropriate tables for the text of the article will increase the value of this review article. (for example, ‘Advantages and disadvantages of the TEE-guided approach to cardioversion’, ‘Recommendation of the use of NOACs in cardioversion’, …etc. )
Author Response
(Reviewer 2)
The article by Fabiana Luca et al. reviewed the latest trend and consensus for ‘the anticoagulation prior to and after cardioversion in patients with atrial fibrillation (AF)’, and also described the recent medical issues of anticoagulation in electrical cardioversion for AF.
This review article is well organized and especially describes the use of non-vitamin K-dependent oral anticoagulants (NOACs) in the setting of cardioversion.
In addition, the authors provide clinical evidence based on a lot of literature, and the review points are of scientific importance.
I have several minor comments:
- The authors used the ‘TOE’ as an abbreviation for transesophageal echocardiography. However, I think that the authors would be better to use ‘TEE’ rather than TOE because the general medical readers might not be familiar with this abbreviation of ‘TOE’.
-We agree with the reviewer. We changed TOE in TEE in the test.
- Page 2, line 90, “Electrical cardioversion in an emergency”: Please provide the anticoagulation strategy after urgent cardioversion for AF
-We agree with the reviewer. We provided the anticoagulation strategy after urgent cardioversion for AF:
After cardioversion a Long Term OAC strategy (OAC for all patients if a CHAD2DS2VASc ≥1 men or ≥2 female) or Short Term OAC strategy (4 weeks OAC if a C HAD2DS2VASc =0 men or ≥1 female (optional if AF onset <24 h) is needed4.
- Please improve the quality of Table 1.
-We agree with the reviewer. We improved the quality of Table
- Finally, the use of appropriate tables for the text of the article will increase the value of this review article. (for example, ‘Advantages and disadvantages of the TEE-guided approach to cardioversion’, ‘Recommendation of the use of NOACs in cardioversion’, …etc. )
-We agree with the reviewer, but in stead of tables, we provided three figures in order to be more immediate by way of illustration. We provided: Figure 1 Decision making on Atrial Fibrillation cardioversion according to 2020 ESC Guidelines; Figure 2 NOAC Therapy in AF CARDIOVERSION NOAC; Figure 3 Transesophageal Echocardiogram in AF Cardioversion
AF, atrial fibrillation; CRNM, clinically relevant non-major; CV, cardioversion; h, hours; LMWH, low molecular weight heparin; ISTH, International Society of Thrombosis and Haemostasis; NOAC, non-vitamin K oral anticoagulant; SE, systemic embolism; TIA, transitory ischemic attack; TEE, transesophageal echocardiography; VKA, vitamin K-antagonist.
Figure 1 Decision making on Atrial Fibrillation cardioversion according to 2020 ESC Guidelines
AF,atrial fibrillation; OAC, oral anticoagulant; NOAC, non-vitamin K antagonist oral anticoagulant;VKA, vitamin K antagonist; UFH, unfractionated heparin; LMWH, low-molecular-weight heparin; TE, thromboembolism; h, hour; CHA2DS2-VASc, Congestive heart failure, Hypertension, Age >_75 years, Diabetes mellitus, Stroke, Vascular disease, Age 65 - 74 years, Sex category (female); m, men; f female; LA, left atrium; LAA, left atrial appendage; TEE, transoesophageal echocardiography.
Figure 2 NOAC Therapy in AF CARDIOVERSION
NOAC, non-vitamin K antagonist oral anticoagulant
Figure 3 Transesophageal Echocardiogram in AF Cardioversion
LAA, left atrial appendage; AF, atrial fibrillation.

Round 2
Reviewer 1 Report
Dear Editors and Authors,
I read improved article entitled 'Anticoagulation in Atrial Fibrillation Cardioversion: What Is Crucial to Take Into Account' with interest.
This review article concerns important topic in cardiovascular medicine: anticoagulation in atrial fibrillation (AF). The general structure of the paper is accurate.
However, I still have some comments to further improve the paper:
- Please explain the abbreviation or do not use it if unnecessary (“the presence of normal velocities in appendage (n.v. 20–40 cm/s)”).
- Please improve grammar/typos: “>40 cm/s has proven proved to be independent predictors of one-year no AF recurrence” or “In stable patients, pharmacological and electrical cardioversion can be performed both. Electrical cardioversion is more effective but, in contrast, sedation is needed” or “apixaban (5 mg if does-adjustment criteria are applied) should be administered. Regarding as the other NAC, a loading dose it has not been advised” or “follow-up is needed lest AF should recurre”.
- Moreover, please write substances beginning from small letters.
- Please explain abbreviations before the first use, e.g. “CKD (Chronic kidney disease)”.
- There are supposed to be 3 figures and a table in the paper but I can not see them in the revised paper. Please submit figures or add them to the paper.
Author Response
- Please explain the abbreviation or do not use it if unnecessary (“the presence of normal velocities in appendage (n.v. 20–40 cm/s)”).
- Please improve grammar/typos: “>40 cm/s has proven proved to be independent predictors of one-year no AF recurrence” or “In stable patients, pharmacological and electrical cardioversion can be performed both. Electrical cardioversion is more effective but, in contrast, sedation is needed” or “apixaban (5 mg if does-adjustment criteria are applied) should be administered. Regarding as the other NAC, a loading dose it has not been advised” or “follow-up is needed lest AF should recurre”.
- Moreover, please write substances beginning from small letters.
- Please explain abbreviations before the first use, e.g. “CKD (Chronic kidney disease)”.
We agree with the reviewer. We corrected the errors.
There are supposed to be 3 figures and a table in the paper but I can not see them in the revised paper. Please submit figures or add them to the paper
We have 3 figures and 2 tables. We uploaded figures and table a zip file.